# SoundSpaces 2.0: A Simulation Platform for Visual-Acoustic Learning

**Changan Chen**[1,4*]  **Carl Schissler**[2*]  **Sanchit Garg**[2*]  **Philip Kobernik**[2]  **Alexander Clegg**[4]
**Paul Calamia**[2]  **Dhruv Batra**[3,4]  **Philip Robinson**[2]  **Kristen Grauman**[1,4]
[1]UT Austin    [2] Reality Labs at Meta    [3]Georgia Tech    [4]Meta AI

## Abstract

We introduce SoundSpaces 2.0, a platform for on-the-fly geometry-based audio rendering for 3D environments. Given a 3D mesh of a real-world environment, SoundSpaces can generate highly realistic acoustics for arbitrary sounds captured from arbitrary microphone locations. Together with existing 3D visual assets, it supports an array of audio-visual research tasks, such as audio-visual navigation, mapping, source localization and separation, and acoustic matching. Compared to existing resources, SoundSpaces 2.0 has the advantages of allowing continuous spatial sampling, generalization to novel environments, and configurable microphone and material properties. To our knowledge, this is the first geometry-based acoustic simulation that offers high fidelity and realism while also being fast enough to use for embodied learning. We showcase the simulator's properties and benchmark its performance against real-world audio measurements. In addition, we demonstrate two downstream tasks—embodied navigation and far-field automatic speech recognition—and highlight sim2real performance for the latter. SoundSpaces 2.0 is publicly available to facilitate wider research for perceptual systems that can both see and hear.[2]

## 1 Introduction

What we see and hear dominates our perceptual experience, and there is often a strong relationship between the two modalities. At the object level, we can anticipate the sounds an object makes based on how it looks, and vice versa (a dog barks, a door slams, a baby cries). At the environment level, materials and geometry of the surrounding 3D space that we see transform the sounds that reach our ears. For example, a person speaking in a marble-floored, high-ceiling museum sounds distinct from one speaking in a cozy carpeted bookshop.

Modeling the correspondence between visuals and acoustics in 3D spaces is of vital importance for many applications in embodied AI and augmented/virtual reality (AR/VR). For instance, a rescue robot needs to localize the person who is calling for help; a service robot needs to look and listen to know if the espresso machine is running properly; an AR system needs to generate sounds that are consistent with the user's acoustical environment for an immersive experience.

Realistic simulations of the first-person perceptual experience are a valuable resource for AI research. They allow training and evaluating models at scale and in a replicable manner. On the visual side, fast visual simulators [60, 70] coupled with 3D assets from scanned real-world environments [11, 78, 68, 56] have facilitated substantial work in visual navigation and related tasks in recent years [77, 12, 6, 30, 57, 34], enabling rigorous benchmarks [1] and even successful "sim2real" transfer to agents that move in the real world [76, 73, 36]. On the audio side, acoustic simulation has been traditionally

---

*Equal contribution
[2]https://github.com/facebookresearch/sound-spaces

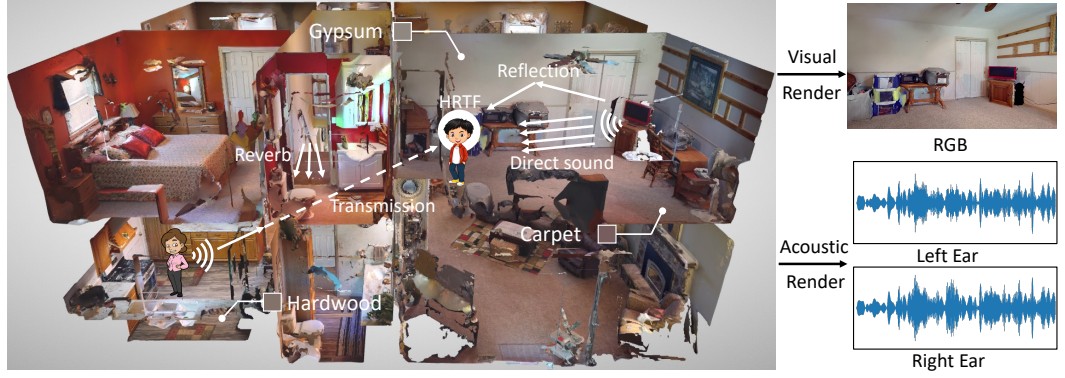

Figure 1: Illustration of SoundSpaces 2.0 rendering in a multi-room multi-floor HM3D [56] environment. In this scenario, a boy is watching TV in the living room while his mom calls him to have dinner from the kitchen downstairs. We model various frequency-dependent acoustic phenomena for sound propagation from all sources (TV and mom) to him, including direct sound, reflection, reverb, transmission, diffraction and air absorption. The sound propagation is based on a bidirectional path-tracing algorithm that takes the geometry of the scene as well as materials of objects in the space as input. The received sound is spatialized to binaural with the head-related transfer function (HRTF). As a result, SoundSpaces 2.0 renders the visual and audio observations with *spatial and acoustic correspondence*. For example, the TV being situated more towards the right results in right-ear signals that are stronger than those in the left ear.

pursued for physical models [9], gaming [43] and auralization for architectural design [75], typically restricted to simple parametric geometries and in isolation from visual context.

Towards bringing the two modalities together in joint audio-visual simulations, recent work offers initial steps [14, 21]. SoundSpaces [14] provides highly realistic room impulse responses (RIRs)[3] obtained via bidirectional path tracing for 100 real multi-room environment meshes from Replica [68] and Matterport3D [11], while ThreeDWorld [21] uses physics simulations in the Unity3D video game platform to model object collisions, impact sounds, and environment reverberation. These tools support an array of new research in navigation [22, 17, 14, 15, 12, 45], floorplan reconstruction [55], feature learning [23], audio-visual (de)reverberation [16, 13], and audio-visual collision detection [20].

Though inspiring, these early platforms have several limitations. SoundSpaces's [14] foremost limitation is its pre-computed, discretized nature. The provided RIRs are pre-computed for all source and receiver pairs on a 0.5m grid, and for a fixed list of 100 total environments. This prevents sampling data at new locations. This in turn means that 1) an agent in the simulator can only move or hop between discrete grid points in the space, which abstracts away some difficult parts of the navigation task; 2) the simulations do not generalize to novel environments—just the 100 provided; and 3) the pre-computed data itself is on the order of TBs, impeding the ability to change configurations, e.g., of the microphone types or materials. ThreeDWorld [21] offers continuous-space rendering, yet it only supports audio rendering for simple 3D environment geometry, namely an oversimplified "shoebox" (rectangular parallelepiped) model, and thus is not applicable to real-scan datasets [78, 56]. In sum, today's audio-visual rendering platforms fall short in accuracy, speed, and flexibility, which in turn constrains the scope of research tasks they can support within audio-visual embodied learning [12, 17, 79] and visual-acoustic learning [67, 44, 13].

In this work, we introduce SoundSpaces 2.0, which performs on-the-fly geometry-based audio rendering for arbitrary environments. It allows highly realistic rendering of arbitrary camera views and arbitrary microphone placements for waveforms of the user's choosing, accounting for all major real-world acoustic factors: direct sounds, early specular/diffuse reflections, reverberation, binaural spatialization, and frequency-dependent effects from materials and air absorption (illustrated in Fig. 1). Furthermore, SoundSpaces 2.0 generalizes audio simulation to *any* input mesh, making it possible for

---

[3]A room impulse response (RIR) is the transfer function that defines how sound is transformed by the environment for a given source and receiver (microphone) location pair.

Table 1: Comparison with existing non-commercial datasets/simulation platforms. *Geometric* refers to acoustic simulation that is based on geometry of the objects and the space. *Configurable* means ability to alter simulation parameters, material and microphone properties. *Arbitrary Env* refers to the ability to render for an arbitrary new mesh environment, including point clouds generated in the wild.

| Platform | Audio-Visual | Geometric | Configurable | Arbitrary Env |
|---|---|---|---|---|
| SoundSpaces [14] | ✓ | ✓ | ✗ | ✗ |
| GWA [71] | ✗ | ✓ | ✓ | ✗ |
| ThreeDWorld [21] | ✓ | ✗ | ✓ | ✗ |
| Pyroomacoustics [61] | ✗ | ✗ | ✓ | ✗ |
| SoundSpaces 2.0 (Ours) | ✓ | ✓ | ✓ | ✓ |

the first time to import sound into well-used environment assets like Gibson [78], HM3D [56], and Matterport3D [11], as well as any future or emerging one like Ego4D [25]. In addition, SoundSpaces 2.0 allows users to configure various properties of the simulation such as source-receiver locations, simulation parameters, material properties, and the microphone configuration. The rendering platform and associated research codebase are publicly available.

In this paper, we describe the new platform and its functionality, and we illustrate its flexibility with various concrete examples (please see also the Supplementary video). In addition, we perform systematic experiments to answer two questions: 1) how accurate are the audio-visual simulations? and 2) how well can machine learning models trained in SoundSpaces 2.0 generalize to real world data? For this purpose, we collect real-world audio RIR measurements for a public scene dataset Replica [68] and benchmark the simulation accuracy. We also benchmark two downstream tasks: continuous audio-visual navigation and far-field speech recognition. For speech recognition, we show the machine-learning models trained on our synthetic data can generalize when tested on real data. We propose an acoustic randomization technique that models the real-world distribution of materials' acoustic properties, and we show that this strategy leads to better sim2real generalization. Finally, aside from the rendering engine itself, which is readily integrated with Habitat [60], we also release SoundSpaces-PanoIR: a large-scale dataset of images paired with RIRs computed in SoundSpaces 2.0; this prepared dataset can facilitate future research on visual-acoustic learning in a stand-alone manner (without interfacing with the simulators themselves).

## 2 Related Work

We overview related work on simulations, audio(-visual) learning, and sim2real transfer.

**Acoustic simulation.** Sounds are first produced by vibrating objects and then propagate in space before reaching human ears. Modeling sound propagation has a long history in the literature, the goal of which is to simulate realistic high-fidelity audio that is consistent with the given environment specification. Interactive acoustic simulation systems have been extensively used in games and AR/VR applications. Sound propagation algorithms typically fall into two main categories: wave-based [3, 35, 50] and geometric [19, 42, 64]. Wave-based methods aim to solve the wave equation numerically, resulting in high computation expense. In the geometric method family, the Image-Source Methods [4] solve the specular reflection of sounds deterministically but have low accuracy for late reverb, while path-tracing based approaches offer both high accuracy and efficiency [59]. Aside from sound propagation, some simulators like TDW [21] model impact sounds between objects.

Our work builds on the SoundSpaces [14] dataset in that we use their bidirectional path-tracing algorithm and simulation framework as a starting point; however, as discussed above, we overcome its core limitations by enabling on-the-fly rendering, and we also augment the propagation algorithm by adding diffraction and improving reverberation level accuracy. Compared to existing public platforms, SoundSpaces 2.0 adds significant generality and flexibility—accepting arbitrary scene geometry, generalizing to new 3D meshes on-the-fly, rendering in real-time, and allowing configuration of materials and microphones—all of which we demonstrate. See Table 1 for comparisons.

**Audio-visual learning.** Recent advances in audio-visual learning include self-supervised cross-modal feature learning from video [5, 40, 49], object localization [28], and audio-visual speech

enhancement and source separation [18, 53, 80, 2, 48, 81, 27]. Besides learning from video, audio-visual simulation supports the study of embodied tasks like navigation [14, 22, 17, 45], where an agent moves intelligently based on the visual and auditory observations. Another line of research facilitated by simulation is visual-acoustic learning [67, 16, 13, 44, 46], where the goal is to either match or remove the room acoustics implied by the image. Acoustic rendering also facilitates audio-only research, such as far-field speech recognition [38, 47, 71], sound source separation [31, 7], localization [26, 31] and sound synthesis [33, 72]. Our work revisits multiple audio(-visual) learning tasks to showcase the advantages of having a continuous, configurable, and generalizable simulation.

**Simulation-to-reality transfer.** Several large-scale datasets of real-world 3D scans of buildings have been released in the past few years [78, 11, 56]. In parallel, multiple simulation environments [78, 60, 39] have been created in order to simulate embodied motion in these 3D scans. These advances allow large-scale training, fast experimentation, consistent benchmarking, and replicable research compared to physical experimentation. Transferring the model trained in simulation to the real world is thus of great interest. The mostly widely used approaches are domain randomization [73, 74], system identification [36, 41], and transfer learning and domain adaptation [82], Most sim2real transfer research studies transferring a policy from simulation to the real world based on visual input. While some work leverages synthetic audio data for speech tasks [71, 31] or builds a multisensory object dataset for sim2real [24], transferring models trained on acoustic simulation has been understudied due to the lack of real-world benchmarks. To the best of our knowledge, this is the first work to both benchmark simulation performance with real measurements (Sec. 5.2) as well conduct sim2real transfer for machine learning models (Sec. 5.4).

## 3 SoundSpaces 2.0 Audio-Visual Rendering Platform

In this section, we detail the audio-visual rendering pipeline for SoundSpaces 2.0.

### 3.1 Rendering Pipeline and Simulation Enhancements

The core of SoundSpaces 2.0 is the audio propagation engine (RLR-Audio-Propagation) we are releasing for research purposes.[4] We integrate this engine into the existing visual simulator Habitat-Sim [60], which offers fast visual rendering.[5] In addition, we provide high-level APIs for various downstream tasks (e.g., navigation) and training scripts at the SoundSpaces repo.[6]

Fig. 1 illustrates the propagation pipeline. SoundSpaces 2.0 takes the scene mesh data processed by Habitat, together with source and receiver locations specified by the user, and computes a room impulse response (RIR) using a bidirectional path-tracing algorithm [10]. This module models various acoustic phenomena, including reflection, transmission, and diffraction, as well as spatialization. The simulation operates in $M$ logarithmically-spaced frequency bands (configurable), where it computes an energy-time histogram at the audio sampling rate. This histogram incorporates spatial information using spherical harmonics for each time sample that represents the directional distribution of arriving sound energy. This representation is then spatialized to either an ambisonic or binaural pressure impulse response [66], which can be convolved with the source audio signals to generate the sound at the receiver position. See Supp. for more details.

Compared to the original SoundSpaces, we have improved the simulation in a few ways. SoundSpaces did not include any simulation of acoustic diffraction, and thus exhibited abrupt occlusion of sources. We have removed this limitation using the fast diffraction approach from [65], which is able to efficiently compute smooth diffraction effects for occluded sources. We also improved the accuracy of the direct-to-reverberant ratio (DRR), the ratio of the sound pressure level of a direct sound from a directional source to the reverberant sound pressure level, by fixing a bias of $\sqrt{4\pi}$ that was present in the indirect sound pressure of the original SoundSpaces.

In the following, we overview modeling advances in SoundSpaces 2.0 that promote continuity, configurability, generalizability, and performance.

---

[4]https://github.com/facebookresearch/rlr-audio-propagation
[5]https://github.com/facebookresearch/habitat-sim/blob/main/docs/AUDIO.md
[6]https://github.com/facebookresearch/sound-spaces

## 3.2 Continuity

**Spatial continuity.** Humans move around in the real world continuously while hearing. Given an arbitrary source location $s$, receiver location $r$, and receiver's heading direction $\theta$ in a given mesh environment, we render the impulse response between the source and receiver as $R(s, r, \theta)$. The sound received by the receiver is computed as $A^r = A^s * R(s, r, \theta)$, where $A^s$ is the sound emitted from the source and $*$ denotes convolution. Whereas SoundSpaces [14] restricts the $s$ and $r$ locations to a 0.5m discrete grid due to its pre-computed approach and hefty storage requirements, SoundSpaces 2.0 allows arbitrary placements.

**Acoustic continuity.** While an agent moves in the environment, it moves smoothly from point A to point B (even with a small step size). With the spatial continuity property, we can render $R(s, r_A, \theta_A)$ and $R(s, r_B, \theta_B)$ for these two locations respectively. However, the original SoundSpaces takes the rendered IR for each location and convolves it with the source sound directly as the audio observation. This calculation implicitly assumes the source does not emit sound continuously, i.e., it starts to emit when the agent moves to a new location, stops after one second, and resumes at the agent's next location.

In SoundSpaces 2.0, we introduce acoustic continuity for both the source sound and listener. More specifically, given a sampling rate $F$ and the time between two steps $\Delta t$, the number of received audio samples is $N = F\Delta t$ per step. Assuming a listener is at location $x_i$ at time $t_i$, the audio signal received by the listener at time $t_i$ emitted from the source at time $t_p$ is $t_i - R(s, x_i, \theta_{x_i}) + 1$. We take the corresponding source sound segment $A^s[t_p : t_p + N]$ and convolve it with $R(s, x_i, \theta_{x_i})$ without zero padding to compute $A_{t_i}^{x_i}$. Following the common practice [51], we apply linear crossfading between $A_{t_i}^{x_{i-1}}$ and $A_{t_i}^{x_i}$ to smooth out the transition from $x_{i-1}$ to $x_i$ with an overlap time window of $T$ seconds. See Supp. video for the impact on perceptual quality.

## 3.3 Configurability

Due to its pre-computed nature, it is impossible to change any simulation setup (parameters, microphones, or materials) for the original SoundSpaces. All are configurable in SoundSpaces 2.0, as summarized below and in more detail in Supp.

**Simulation parameters.** We expose many useful parameters for users to configure, including the sampling rate, the number of frequency bands, number of rays for direct/indirect sounds, whether reflection, transmission or diffraction is enabled, etc.

**Microphone types.** We provide several types of built-in microphone configurations, including monaural single-channel audio, binaural (modeling a human listener), and ambisonics (full sphere surround sound). In addition, users are also able to configure their own microphone array by specifying an array of monaural microphone locations.

**Custom HRTFs.** We allow users to load their own head-related transfer functions (HRTFs), which incorporate customized human perception in the acoustic rendering simulation.

**Material modeling.** Materials of objects/surfaces have a big impact on how humans perceive the sound in an environment. Consider the difference between sound in a recording studio versus a living room of the same size. Due to the absorptive materials in the recording studio, the sound will consist primarily of direct sound without reverberation, whereas in the living room, the sound will consist of a mixture of direct sound and reverberation.

Existing real-scan datasets have semantic annotations at the level of object categories, e.g., chair, table, couch and floor, while lacking material annotations of what these objects are made of, e.g., wood or steel for tables. SoundSpaces coped with this issue by defining a fixed mapping from object categories to acoustic materials, e.g., floors are always mapped to the carpet material, which is very absorptive. However, this fixed mapping fails to reflect the fact in the real world, different instances of the same object category could have very different acoustic properties, e.g., a floor could be carpet or wood or concrete materials depending on the home type.

To account for this variation, we expose an API to let users define their own acoustic material configurations. We provide 29 built-in acoustic materials, e.g., wood, concrete, curtain, soil, water. Every

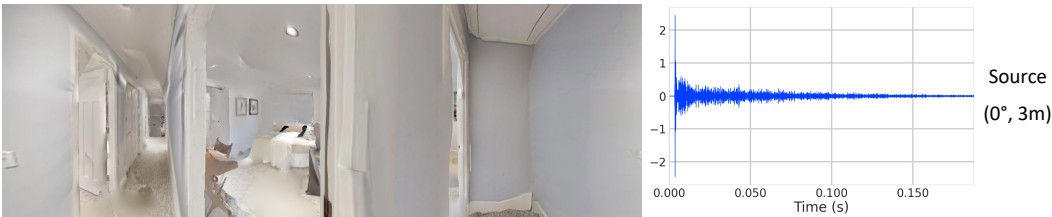

Figure 2: An example of the SoundSpaces-PanoIR dataset rendered on Gibson [78]. We render panoramic RGB and depth images to capture the scene geometry. We provide the images, impulse responses, and the coordinates of the point source location with respect to the current camera pose.

acoustic material has a list of candidate object categories to be mapped from. It also has a set of coefficients for absorption, scattering, and transmission in the following format: $[f_1, c_1, f_2, c_2, ..., f_n, c_n]$, where $f_i$ is a frequency and $c_i$ is the coefficient for a certain acoustic phenomenon at frequency $f_i$. This allows modeling the frequency-dependent acoustic properties of different acoustic materials. For example, high-frequency waves are absorbed more compared to low frequencies when reflecting from carpets. See Supp. for details.

We also model distance-dependent damping of the sound propagation media. This includes air absorption as well as transmission losses through materials. Air absorption is calculated using an analytical model [8]. Users can specify the frequency-dependent damping coefficients for each material, expressed as dB per meter, in a similar format to the other material properties.

### 3.4 Generalizability

**Generalization to scene datasets.** Our new simulator accommodates arbitrary 3D meshes as input. This makes it compatible with all available scene datasets (e.g., Gibson [78], HM3D [56], Ego4D [25], Matterport3D [11], Replica [68]), as well as any future assets that become available, such as if a user scans their own lab or home environment. This is an important advance over SoundSpaces, which was restricted to Replica and Matterport3D alone. See Supp. for videos of generated examples.

**Generalization to shoebox rooms.** We expose APIs for creating shoebox rooms with different materials for walls, which simulates simpler setups as in Pyroomacoustics [61] and TDW [21].

**Generalization to the real world.** The fidelity and flexibility of our simulation platform also supports generalization to the real world. In Sec. 5, we score the simulator output against real-world RIRs and show how machine learning models trained on SoundSpaces 2.0 can generalize to real data.

### 3.5 Rendering Modes and Rendering Performance

Our simulation generates high-quality audio rendering based on mesh and materials, and this fidelity can be instrumental for certain research areas. On the other hand, in tasks like embodied navigation with reinforcement learning, which typically require millions (or even billions [77]) of training iterations, rendering speed is of vital importance. Thus, we offer two built-in rendering modes: *high-speed* and *high-quality*.

In high-speed mode, we reduce the number of rays and improve the accuracy by leveraging previously computed impulse responses [63], under the assumption that movements are spatially continuous. Our algorithms use information computed on previous simulation frames, such as sound propagation paths and RIRs, to reduce the number of rays and ray bounces that are needed on each frame for sufficient sound quality (see Sec.5.1). In high-quality mode, we set all rendering parameters to max and turn off the temporal coherence feature to ensure that every impulse response is accurate without temporal blurring. Our engine is multi-threaded and users can set the number of threads when using either mode. See Sec. 5.1 for analysis of the simulation performance in terms of speed and accuracy.

Table 2: Simulation speed vs. quality tradeoff. We report mean and standard deviation over 5 runs.

| | Relative RT60 Error (%) | 1 Thread (FPS) | 5 Threads (FPS) |
|---|---|---|---|
| High-quality | $0.0 \pm 0.0$ | $0.9 \pm 0.0$ | $4.0 \pm 0.1$ |
| High-speed | $9.5 \pm 0.2$ | $7.7 \pm 0.2$ | $33.5 \pm 0.4$ |

## 4 Large-scale SoundSpaces-PanoIR Dataset

We are releasing SoundSpaces 2.0 as a general-purpose platform with which a user can generate observations on-the-fly (particularly relevant for embodied AI models), or populate a new offline dataset of their own design. As an example of the latter, and to ease adoption for researchers who wish to work with visual-acoustic scene data without a layer of agent interaction, we next use SoundSpaces 2.0 to compose a large-scale dataset called *SoundSpaces-PanoIR* that couples IRs with images.

For visual-acoustic learning tasks, such as audio-visual dereverberation [16] and synthesizing acoustics based on visuals [13, 67, 44], there are no existing large-scale accurate image-IR datasets due to the high expense and complexity of data collection. Our SoundSpaces-PanoIR dataset has 10M panoramic image and IR pairs rendered from 750 environments across the Matterport3D, Gibson, and HM3D datasets. We provide the data in the following format: panorama (RGB/Depth), IR, polar coordinates of the source with respect of the center of the panorama. Fig. 2 shows one example in Gibson. See Supp. for more examples and statistics.

## 5 Evaluation and Benchmarks

Next we evaluate both the simulation quality and its value for downstream tasks with two machine learning benchmarks. Fig. 3a illustrates these two tasks.

### 5.1 Simulation Speed vs. Quality Tradeoff

To understand the tradeoff between the quality versus speed of rendering, we report the accuracy and speed of different modes by rendering RIRs along random trajectories with an average length of 15m across 20 Matterport3D environments. We profile the speed on a Xeon(R) Gold 6230 CPU with 2.10GHz. See Table 2. For accuracy, we measure the relative RT60 error of RIRs generated in high-speed mode compared to RIRs generated in high-quality mode. RT60 is a standard acoustic measurement that is defined as the time it takes for the sound pressure level to reduce by 60 dB [29]. We see high-speed greatly improves efficiency over the high-quality mode, by $8\times$ with single thread and $33\times$ with 5 threads, while only losing $9.5\%$ accuracy despite RT60 calculation being noisy. When coupled with distributed training, it meets the requirement of today's RL agent training. In addition, we test the navigation model trained in high-speed mode on high-quality mode; the performance difference is smaller than $1\%$ compared to the test performance in high-speed mode in Table 3. In comparison, TDW [21] runs at 60 FPS and SoundSpaces runs at 500+ FPS (bottleneck on I/O) at the cost of simplified room models or not being configurable, respectively, c.f. Table 1. We treat high-quality mode as the gold-standard and benchmark its quality against real-world IRs next.

### 5.2 Validating Simulation Accuracy with Real IRs

How realistic are our audio simulations? To quantify this, we collect real acoustic measurements of the FRL apartment from the Replica dataset [68] and compare them to SoundSpaces 2.0 outputs. IR measurements were captured at seven different source/receiver positions throughout the real-world apartment using an omnidirectional B&K Type 4295 speaker (100Hz to 8kHz frequency response) and Earthworks M30 microphone with the exponential sine sweep method. These measurements are publicly available to assist future research.

Figure 3b compares the measurements to the corresponding simulations at the same source/receiver positions, for both the original SoundSpaces and the proposed SoundSpaces 2.0 (high-quality mode). [7]

---

[7]The measurements were scaled to match the direct sound level of the simulations. The acoustic material properties of the mesh were optimized to match the measurements following [62].

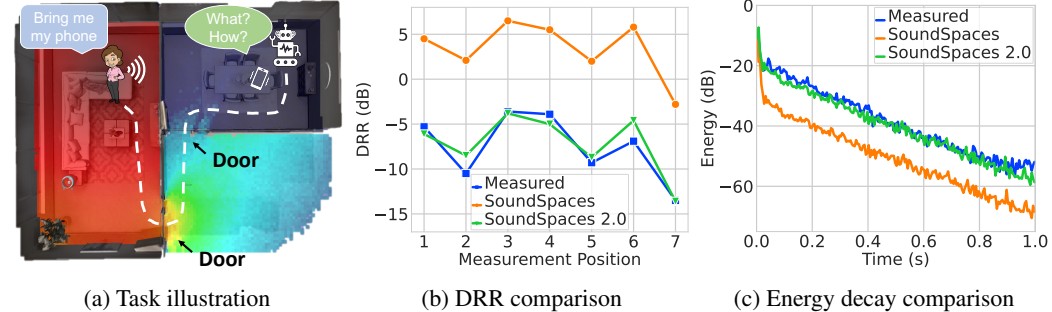

|     (a) Task illustration     |     (b) DRR comparison     |     (c) Energy decay comparison     |

Figure 3: **(a)** In this example, a person's phone rings in the dining room while she is in the living room and she asks the robot to bring her the phone. Upon receiving the audio signal with the binaural microphone, the robot needs to figure out two things: 1) what she is saying (far-field automatic speech recognition) and 2) how to navigate to her and the phone (audio-visual navigation). Note that far-field ASR is not limited to robotics; it has various applications such as video captioning. **(b)** Comparing real measurements and simulations in the Replica apartment [68] for 7 measurement positions and the 250Hz to 4000Hz frequency band. SoundSpaces 2.0 has much lower error for the direct-to-reverberant ratio (DRR) compared to SoundSpaces. **(c)** Energy decay curve comparisons. The energy decay curve of SoundSpaces 2.0 is much closer to the real measurements than SoundSpaces.

Table 3: Continuous audio-visual navigation benchmark. DTG stands for distance to goal. We report the mean and standard deviation by training on 1 random seed, and evaluating on 3 random seeds.

| Train | Test | Success (%) | SPL (%) | DTG (m) |
|---|---|---|---|---|
| SoundSpaces [14] | Continuous space | $64.2 \pm 0.8$ | $27.5 \pm 0.4$ | $5.6 \pm 0.2$ |
| SoundSpaces [14] | Continuous space & continuous sound | $0.9 \pm 0.2$ | $0.3 \pm 0.1$ | $12.9 \pm 0.1$ |
| SoundSpaces 2.0 | Continuous space & continuous sound | $64.7 \pm 3.9$ | $49.3 \pm 3.0$ | $5.9 \pm 0.5$ |

We report the direct-to-reverberant ratio (DRR) acoustic parameters derived from the impulse responses [29] in Figure 3b. SoundSpaces 2.0 has a better match of direct-to-reverberant ratio, where the error compared to measurements is reduced from 11.0 dB to 0.98 dB on average, while preserving the same relative RT60 error of 12.4% (see Supp.). Figure 3c reinforces that advantage, plotting the energy-time curves of the simulations versus the real measurements from 250Hz to 4000Hz. Overall, the proposed new features and improvements lead to higher realism for the acoustic simulation.

### 5.3 Benchmark 1: Continuous Audio-Visual Navigation

Navigating to localize the sound source in an unmapped environment has many real world applications, such as rescue robot or service robot (e.g., find the person calling for help or the ringing phone). The audio-visual navigation task (AV-Nav), originally introduced in [14], is gaining attention from the broader community via public competitions at CVPR 2021 and CVPR 2022. [8] However, due to its reliance on SoundSpaces, AV-Nav thus far must assume the agent travels along the discrete grid. The navigation task is thus easier due to the lack of collisions and implied perfect localization.

Here we introduce the *continuous* AV-Nav task, enabled by SoundSpaces 2.0 simulation. In this task, the agent can either move forward 0.15 m per step at a speed of 1m/s or turn left/right 10 degrees. If the agent issues a stop action within 1m radius of the goal, the episode is regarded as successful. Importantly, the agent not only moves in continuous space but also receives acoustically continuous audio signals (cf. Sec. 3.2). We use the high-speed rendering mode.

We generalize the existing audio-visual navigation (AV-Nav) agent [14] to a distributed audio-visual navigation (DAV-Nav) agent equipped with DD-PPO [77] to speed up the training process. We train and test on the AudioGoal navigation dataset [14]. To ablate the simulation improvement as detailed in Sec. 3.1, for the SoundSpaces baseline, we train DAV-Nav on SoundSpaces' discrete setup (agent only moving between grid points) with data rendered from the enhanced simulation; the action space is either moving forward 1 m, turning left/right 90 degrees or issuing a stop action.

---

[8] https://soundspaces.org/challenge

Table 4: Far-field automatic speech recognition benchmark.

|  | Word Error Rate (%) |
| --- | --- |
| Pretrained | 29.10 |
| Finetuned on real IRs [38] | 13.32 |
| Finetuned on Pyroomaoustics [61] | 16.24 |
| Finetuned on SoundSpaces 1.0 [14] | 18.48 |
| Finetuned on SoundSpaces 2.0 | **12.48** |

Table 3 shows the results using the standard metrics of success rate, success rate normalized by path length (SPL), and distance to goal. If only the space is continuous, the DAV-Nav agent trained on SoundSpaces has 64.2% success rate and 27.5% SPL on average compared to 64.7% success rate and 49.3% SPL of the agent trained on SoundSpaces 2.0. This shows spatial continuity mostly harms the agent's efficiency rather than its success rate; the agent can still navigate to the source despite having more collisions. However, when the sound is acoustically continuous, the baseline's performance drops. This is likely because the agent relies on the direct-sound cue that is (inaccurately) always present in the audio, while in the continuous-sound rendering, direct sound is always mixed with the reverberation in the environment, making navigation more difficult. In comparison, the agent trained on SoundSpaces 2.0 achieves a much higher success rate and is much closer to the goal location on average. This shows it is essential to model both spatial and acoustic continuity for AV-Nav, which SoundSpaces 2.0 enables. Furthermore, recall that SoundSpaces 2.0 opens up any other 3D scene dataset for exploring AV-Nav, whereas previously only Replica or Matterport3D were applicable.

## 5.4   Benchmark 2: Far-Field Automatic Speech Recognition

Speech recognition is critical for many applications, including far-field scenarios where the speaker is far from the microphone (e.g., speaking to a smart home assistant device). When speech recognition models are trained on a clean speech corpus, such as LibriSpeech [54], they generalize poorly to far-field cases with unanticipated reverberation. Due to the high expense of collecting real IRs, synthetic impulse responses are thus often used to augment speech for far-field ASR [38, 47, 71]. Here, we propose to benchmark far-field ASR systems augmented by our generated impulse responses.

We take the pretrained transformer-based ASR system from SpeechBrain [58], an open-sourced speech toolkit, as the base model. For finetuning, we augment speech in the train-clean-100 split of LibriSpeech [54] with IRs generated in different systems and finetune 60 epochs. For testing, we augment speech from a real RIR dataset [69], where IRs are recorded in real environments, e.g., home, conference rooms, auditoriums. In this way, we test the sim2real generalization for models trained on the synthetic data. We compare the pretrained model with the ASR model finetuned on IRs generated with Pyroomacoustics, SoundSpaces 1.0, and SoundSpaces 2.0 (high-quality mode). We ensured the simulated RIRs have matching RT60 distributions. In addition, we compare with the ASR model finetuned on real IRs [38] from the RWCP sound scene database [52], the 2014 REVERB challenge database [37], and the Aachen impulse response database (AIR) [32].

Table 4 shows the results. As we can see, the pretrained model generalizes poorly to far-field speech with word error rate (WER) of 29.1%, compared to 2.4% WER on a clean test set lacking any reverberation. Finetuning with synthetic IRs leads to a dramatic improvement. Comparing Pyroomacoustics and SoundSpaces 1.0 to SoundSpaces 2.0, our generated IRs lead to much lower WER. Finetuning on real IRs also reduces the error substantially, but still not as much as our simulated data, which can be generated at scale across a wide variety of environments. Our simulation generates realistic IRs that help machine learning models generalize better to reality.

**Acoustic randomization.**   In the real world, instances of a given object category need not share identical material profiles. While existing simulations do not model such nuances, in SoundSpaces 2.0 we can manipulate the materials in a more subtle way. Inspired by domain randomization techniques [73, 74] that randomize simulation parameters for better sim2real generalization, we explore if acoustic randomization offers similar benefits. Specifically, we define a set of possible acoustic materials for each object category. When rendering, a random material is picked for a category to simulate the category-level variation. In addition, to model the differences of acoustic

materials, we add $\mathcal{N}(0, 0.1)$ Gaussian noise to each coefficient. Altogether, this strategy models both the category-level and instance-level material nuances.

When we use the proposed acoustic randomization technique to generate the same amount of data for finetuning, the ASR model has even lower WER on the test set, reduced from 12.48% to 12.04%, while uniform randomization, i.e., uniformly sampling coefficients between 0 and 1, leads to a higher WER of 12.58%. This not only shows the benefit of acoustic randomization but also how SoundSpaces 2.0's configurability facilitates research on acoustic sim2real.

## 6    Discussion on Limitations and Future Work

We believe SoundSpaces 2.0 can facilitate significant new work in embodied AI, multimodal perception, and audio research. The platform is general and accessible, and our experiments offer concrete examples of its potential.

Like any research tool, there are certain limitations and assumptions that are important to recognize. Our simulation platform supports audio rendering for arbitrary environments with a state-of-the-art path tracing algorithm. For this algorithm to render accurately, the scene meshes need to have high quality, i.e., no large open holes on the mesh, otherwise the rays will leak from the holes, resulting in inaccurate simulation. For example, Section 5 in Supp. shows Matteport3D has lower RT60s compared to other datasets on average due to some of its broken mesh environments. To aid users in checking the mesh quality for audio rendering, we expose an API to let users check the percentage of rays leaked from the mesh; users can repair the mesh accordingly if the ray efficiency is low. Path tracing is also vulnerable to the standard shortcomings of geometrical-acoustics techniques, e.g., room modes, though our implementation takes care to eliminate the typical lack of diffraction as described in Sec. 3.

Materials have impact on audio simulation, and one of the open challenges of material modeling is that it is infeasible to accurately know the acoustic material properties only given the environment meshes, e.g., we cannot estimate how much energy the floor absorbs purely based on the mesh or rendered visuals. Currently, we tackle that by assigning common material properties to objects (Sec. 3.3), which allows our simulator to operate with fairly lightweight assumptions about the incoming mesh. For more in-depth treatment of materials, one could perform acoustic measurements into the environment scanning pipeline when creating a digital replica of a real-world environment.

In this work, we validate the simulation accuracy with real IRs collected in the apartment from the Replica dataset. To improve the simulation accuracy and further understand its difference with the real world, future work could collect acoustic measurements in diverse environments with varying geometry and materials, which is supported by our simulation platform (Sec. 3.4).

## 7    Conclusion

We introduced SoundSpaces 2.0, a platform for on-the-fly geometry-based audio rendering for 3D environments. We collected real measurements in public 3D scenes to validate the simulation accuracy. We benchmarked two tasks and showed encouraging evidence that systems trained on this simulation can generalize to the real world. Notably, beyond those two tasks, our platform will supportand upgrade many other tasks currently explored in the literature [12, 17, 44, 31, 55, 67]. Lastly, we are releasing a large-scale SoundSpaces-PanoIR dataset that is ready to use for research. We have made the simulation and real measurements publicly available to facilitate research on visual-acoustic learning.

**Acknowledgements:** UT Austin is supported in part by a gift from Google, DARPA L2M, and the IFML NSF AI Institute.

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
