# OpenReview forum: "SoundSpaces 2.0: A Simulation Platform for Visual-Acoustic Learning"
_NeurIPS.cc/2022/Track/Datasets_and_Benchmarks — NeurIPS 2022 Datasets and Benchmarks _

### Official Review · Reviewer_oZ85 · 2022-07-21
**Significant infrastructure for visual-acoustic learning; room for improvement for experimental evaluation**

**Rating:** 7
**Confidence:** 4

**Strengths:**

The paper proposes a solid, open-sourced implementation of an audio simulation environment that performs on-the-fly geometry-based audio rendering for arbitrary scenes. It’s first of its kind and can open up many research opportunities for (multi-modal) embodied AI. In terms of accessibility, everything seems to be open-sourced. The platform supports two modes - high-speed and high-quality, and also includes a large-scale offline dataset called SoundSpaces-PanoIR if the users are not interested in embodied AI tasks.


**Weaknesses:**

The experimental evaluation section has a few noteworthy weaknesses.

First, it’s a bit unclear why continuous audio simulation is important. Intuitively, it’s better because it’s closer to the real world. But the experimental results (Table 3 specifically) are not very convincing. Comparing row 2 and row 3, it’s expected that training in SoundSpaces and testing in SoundSpaces 2.0 will cause a large domain shift, which is known to be very detrimental for deep learning models. It’s still unclear whether having continuous audio simulation is important, if the ultimate goal is to deploy a home service robot in the real world (a motivation that is repeatedly mentioned in the paper).

The results of Table 4 is more convincing: clearly, SoundSpaces 2.0 generates more realistic synthetic IRs than Pyroomaoustics for the purpose of ASR. However, table 4 should also include the result of “Finetuned on SoundSpaces”. In addition, figure 3 (b) is a bit suspicious. There seems to be a constant delta between the results of SoundSpaces and SoundSpaces 2.0. Where is that delta coming from? In other words, why is SoundSpace always producing DRR 10 dB higher?

Furthermore, in table 2, the authors should probably also compare with SoundSpaces (1.0) and ThreeDWorld. They have certain simplifications (discrete grid point for SoundSpaces, and “shoebox” simplification for TDW), and presumably, they can run faster. The users should be aware of the full spectrum of the simulation speed versus quality tradeoff when choosing which simulator to use. For example, if some user needs >100 FPS or >1000 FPS for their RL use case, maybe they will choose SoundSpaces over SoundSpaces 2.0.

Finally, there seems to be missing a clear comparison with ThreeDWorld in the experimental evaluation section. ThreeDWorld also supports real-time, continuous audio rendering and also impact sound simulation, which is missing from SoundSpaces 2.0. The main limitation seems to be its assumption of simplified scene geometry (“shoebox”). The authors should demonstrate how this limitation affects the learning of downstream tasks, including embodied ones (e.g. audio-visual navigation) or non-embodied ones (e.g. ASR).


**Additional Feedback:**

In the supplemental video, the section of comparison with real measurements: SoundSpaces 2.0 is definitely better than SoundsSpaces, but it’s still quite different from the real IR. What causes the gap and how you would propose to fill the gap in the future?

Minor typo in Fig 3 caption: It should be “(b) Comparing real measurements and simulation”, not “(c) Comparing…”


**Clarity:**

The paper is clearly written.


**Correctness:**

The simulation environment and the dataset seem to be constructed in a very sound way. The paper proposes two example downstream tasks that can leverage the simulator: AV-Nav and ASR, although the simulator can probably unlock more research directions than what is demonstrated.


**Documentation:**

The authors listed all the relevant URLs in the paper. Everything seems to be open-sourced under CC-BY-4.0, CC-BY-NC and MIT licenses.


**Ethics:**

No ethical concerns were found.

**Relation To Prior Work:**

The paper has a pretty comprehensive related work section that covers three aspects: acoustic simulation, audio-visual learning, and sim-to-real transfer. In the acoustic simulation section, it’s probably advisable to include ThreeDWorld (TDW is discussed in the introduction, but not in related work). It would also be better to explicitly acknowledge here that TDW simulates impact sounds of object collision, whereas SoundsSpaces 2.0 cannot. Finally, in the sim2real transfer section, the authors missed the citation of ObjectFolder 2.0 [1], which indeed conducted sim2real transfer of machine learning models (despite in slightly different scenarios, but still quite related)

[1] Gao, Ruohan, et al. "ObjectFolder 2.0: A Multisensory Object Dataset for Sim2Real Transfer." Proceedings of the IEEE/CVF Conference on Computer Vision and Pattern Recognition. 2022.


**Summary And Contributions:**

The paper introduces SoundSpaces 2.0, the first geometry-based real-time acoustic simulation platform for 3D environments. The main difference from its previous version (SoundSpaces) is its continuous acoustic simulation (as opposed to discrete), which brings three benefits: 1) more accurate acoustic simulation, not restricted to predefined grid points, 2) generalization to new scenes (arbitrary input meshes), 3) better configurability (microphones, materials, etc). In terms of experimental evaluation, the authors compare the simulated sounds with real-world audio measurements. They also showcase two downstream tasks that leverage the simulator: audio-visual navigation and far-field automatic speech recognition.

---

> ### Author Response · Authors · 2022-08-29
> **Response to reviewer oZ85**
>
> We thank the reviewer for the encouraging feedback and insightful questions.
>
> **Q1: Why is continuous audio simulation important?**
> Compared to SoundSpaces 1.0, SoundSpaces 2.0 is both acoustically and spatially continuous. In Sec. 5.3, we study how acoustic continuity and spatial continuity affect navigation. And we show that spatial continuity does not have a huge impact on the navigation success rate (row 1 vs row 3), while the assumption of periodic sound (no acoustic continuity) results in most of the drop in the success rate (row 2 vs row 3). This experiment highlights the importance of a continuous simulation environment for sim2real transfer especially where acoustics are continuous.
>
> **Q2: Table 4 should also include ''finetuned on SoundSpaces".**
> Thank you, added.
>
> **Q3: Constant delta in figure 3 (b).**
> This delta mostly comes from the attenuation bias present in the original SoundSpaces (discussed in section 3.1), which results in an incorrect direct-to-reverberant ratio. The rest of the gain comes from other improvements in the simulation accuracy, e.g., the diffraction.
>
> **Q4: More comparisons in Table 2.**
> We can't run the accuracy comparison for SoundSpaces 1.0 and TDW in Table 2 because they can't access the same setup (environments and sampled sound locations). For speed comparison, we have added some in-text analysis about different simulators' speeds in section 5.1.
>
> **Q5: Comparison with TDW.**
> The acoustic simulation used in TDW has the same assumption and approach as Pyroomacoustics, and thus we use Pyroomacoustics to represent this family of approaches. For this family of approach, the direct limitation is that they can not accurately render sound for environments with complex geometry, e.g., scans of real-world houses or rooms with furniture, and thus we can't compare directly with them in a navigation task. However, in the ASR task, we have shown that machine learning models trained on such data do not generalize to real data as well as models trained on our simulated data (Section 5.4, Table 4).
>
> **Q6: Suggested reference.**
> Thank you for the Gao et al. ObjectFolder CVPR 2022 reference, which we have added. TDW is cited and compared in the related work section (Table 1). Note that sound propagation and impact sound simulation are two lines of related work, and the acoustic simulation paragraph in related work is dedicated to cover sound propagation due to the limit of space and relevance. We have acknowledged our lack of TDW's impact sound simulation in Related Work as well.
>
> **Q7: How to fill the gap with real IR in the future?**
> One of the open challenges is that it's infeasible to accurately know the acoustic material properties only given the mesh environments, e.g., currently we can't estimate how much energy the floor absorbs purely based on the mesh or rendered visuals. This calls for incorporating acoustic measurements into the environment scanning pipeline when creating a digital replica of a real-world environment. We have added this discussion in Section 6.

---

### Official Review · Reviewer_fgxG · 2022-07-25
**Review for SoundSpaces 2.0: A Simulation Platform for Visual-Acoustic Learning**

**Rating:** 7
**Confidence:** 3
**Correctness:** The experiment design is appropriate …
**Clarity:** The paper is well and clearly written.

**Strengths:**

1. Novelty. The proposed platform for continuous spatial sampling is novel, and the ideas are original.
2. Detail pipeline. The authors provide the audio-visual rendering pipeline in detail and illustrate the heavy workload.
3. Experiments. The authors compare and benchmark two downstream tasks: continuous audio-visual navigation and far-field speech recognition, and further provide insights on designing these systems.
4. Good presentation. The paper is clearly constructed and presented.

**Weaknesses:**

1. Though two downstream tasks have been benchmarked, it could be better to provide results on extended tasks, such as cross-modal learning and more audio-only research.
2. It's better to provide statistics of the proposed dataset, including evaluation and visualization of the duration, number of speakers, and so on.

**Additional Feedback:**

N/A

**Documentation:**

The authors provide clear detail on data collection and organization. The dataset is publicly available.

**Ethics:**

No ethical concerns.

**Relation To Prior Work:**

SoundSpaces 1.0’s foremost limitation is its pre-computed, discretized nature. In contrast, SoundSpaces 2.0 performs on-the-fly geometry-based audio rendering for arbitrary environments. It allows continuous spatial sampling and generalization to novel environments

**Summary And Contributions:**

Summary:
The authors introduce SoundSpaces 2.0, a platform for on-the-fly geometry-based audio 2 rendering for 3D environments. It allows continuous spatial sampling and demonstrates the effectiveness in two downstream tasks (embodied navigation and far-field automatic speech recognition).

---

> ### Author Response · Authors · 2022-08-29
> **Response to reviewer fgxG**
>
> We thank the reviewer for the encouraging feedback and insightful questions.
>
> **Q1: More downstream tasks and benchmarks.**
> We chose to cover two central tasks in the research area. Exploring even more tasks will indeed be interesting future work; we believe SoundSpaces is a powerful tool  well-positioned to enable such work in the broader community.
>
> **Q2: Statistics of the proposed dataset.**
> The statistics of PanoIR dataset are covered in Sec. 8.5 in the supplementary material.

---

### Official Review · Reviewer_DktZ · 2022-07-25
**A new embodied AI platform simulating sound**

**Rating:** 7
**Confidence:** 4
**Correctness:** Yes
**Clarity:** For the most part.

**Strengths:**

An upgraded version of the platform overcomes some major limitations of the prior work.
Some experiments demonstrate that continuous inputs do impact model performance.
Provided datasets and platforms can be quite useful to the community.

**Weaknesses:**

Details of the sound sources are not clear. An object might drop and only produces one impact sound.  Not sure if the current platform supports this natural interaction.

For experiment 2, the authors are using high-quality mode. It's interesting to see how the faster version compared to the slower version. So we can better understand the impact of this trade-off.

Overall it’s a good paper. However, the authors should include more details to reach a broader audience and to help the community understand what can be done with the proposed platform. A more detailed limitation section or some comparison with real-world sound sources should help.


**Additional Feedback:**

N/A

**Documentation:**

There is a lack of detail in the dataset collection process In the paper. The paper mainly talked about the platform.

**Relation To Prior Work:**

Very clear.

**Summary And Contributions:**

This paper proposed a new simulator for simulating acoustic effects. The major difference compared to the prior work is that this work can compute sound properties on the fly and thus avoid using precomputed data. Therefore, the new platform can support arbitrary geometry and continuous outputs. The authors also perform two experiments to provide evidence for the usefulness of this work

---

> ### Author Response · Authors · 2022-08-29
> **Response to reviewer DktZ**
>
> We thank the reviewer for the encouraging feedback and insightful questions.
>
> **Q1: Details of sound sources. What if an object drops?**
> There is no constraint on the content of the sound sources. This is feasible because sound propagation is linear with a fixed configuration of the environments and source/listener locations. Users can insert arbitrary sounds of their choice at any location at run time, and the output sound at the target location is the convolution of the sound with the rendered impulse response.
>
> As for modeling an object dropping in the environment: the embodiment system Habitat we build upon supports interactions between objects, but impact sound modeling is not currently incorporated into our pipeline.  We added an emphasis on this point in the Related Work.
>
>
> **Q2: High-quality mode in experiment 2.**
> In Section 5.1, we show that the high-speed mode's RT60s deviate no more than 10\% compared to high-quality mode, and when the data is used by downstream tasks, the performance difference of ML models is within 1\%.
>
> **Q3: More details to reach a broader audience and more detailed limitation section.**
> The limitation of this work was initially discussed in the Supp. material. With the additional page available at rebuttal, we pushed this content to the main in our revision. See also our responses to Reviewer TAC1.
>
> **Q4: Lack of detail in the dataset collection process.**
> Please see Supp. 8.5 for the details of the dataset collection process (ref in L244).

---

### Official Review · Reviewer_JZjD · 2022-07-27
**Solid contribution to visual-acoustic simulation**

**Rating:** 7
**Confidence:** 4
**Clarity:** Yes, the writing is good.

**Strengths:**

The contributions are significant. First, the simulation now supports real-time, continuous audio rendering; second, the rendering is also very realistic based on bidirectional path tracing; third, the simulation is now configurable and can be used as a general tool to simulate any 3D environment. All these features are much needed for embodied AI research.

The dataset and the benchmarking results are also good to have for the community.


**Weaknesses:**

I see two weaknesses regarding the submission.  First, the methods used in benchmarking are limited. The authors used one algorithm for each of the two benchmark tasks.  The one for navigation is also introduced in the previous version of the work.  It'd be good to see a more systematic evaluation with additional methods.

Second, the sim-to-real experiments can be strengthened.  The authors conducted some preliminary experiments in Benchmark 2.  It'd be great to see more of these experiments in more benchmarks and tasks to show that not only the simulation itself is realistic, but models trained in these environments do transfer to the real world.

**Additional Feedback:**

None.

**Correctness:**

Mostly yes. However, the methods considered in the evaluation are limited, and the simulation-to-real experiment is mostly a proof-of-concept and can be strengthened.

**Documentation:**

Yes.

**Ethics:**

No concerns.

**Relation To Prior Work:**

Yes, the related work section is complete.

**Summary And Contributions:**

In this submission, the authors introduced SoundSpaces 2.0, a simulation platform that supports audio simulation with a number of improvements over existing platforms. In addition, the authors have validated the simulation accuracies with real-world recordings, introduced a dataset based on the environment, and conducted benchmarking on two tasks.

---

> ### Author Response · Authors · 2022-08-29
> **Response to reviewer JZjD**
>
> We thank the reviewer for the encouraging feedback and insightful questions.
>
> **Q1: More systematic evaluation with additional methods.**
> We'd welcome more details here as far as what method the reviewer finds is missing. For benchmarking audio-visual navigation, we would argue our proposed DAV-Nav approach is a strong baseline that is hard to beat. We adapt the existing AV-Nav method to the continuous audio-visual navigation task with the decentralized distributed proximal policy optimization (DDPPO), which has been shown very effective with sufficiently large amount of training samples (80 million in our case). For the ASR experiment, Pyroomacoustics is the most commonly used open-source software for room acoustics simulation (no visual simulation though). As requested by other reviewers, we added two additional baselines in the ASR experiment (Table 4) and our model still outperforms both.
>
> **Q2: How do models trained transfer to the real world?**
> To validate the realism of our simulation, we not only compared our simulation with real measurements (Section 5.2) but also transferred machine learning models trained on our simulation to the real world (real IR dataset) in Section 5.4.  We show that ML models trained on our simulated data generalize better compared to models trained on another acoustic simulation package PyRoomacoustics, as well as on real IRs.  Exploring more sim2real tasks will indeed be interesting future work, and we believe our SoundSpaces 2.0 data generation platform is a powerful tool set up exactly to kickstart such work in the broader community.

---

### Official Review · Reviewer_TAC1 · 2022-07-28
**Although the claims to realism are not as strong as I would like, this is nonetheless an useful tool**

**Rating:** 7
**Confidence:** 4

**Strengths:**

The codebase presented here provides a useful tool for synthesizing acoustic reverberation using mesh-grids formatted for common scene simulation tools. The codebase improves upon its first generation (soundscape) by allowing continuous sampling of locations for the sound source and receiver. This almost certainly results in more accurate changes of sound with listener location, which is plausibly why simulated agents can better localize sources in such an environment.  Spatial audio in VR environments may well also benefit from this update.

The authors have provided evidence of sim2real transfer for a speech recognition algorithm trained on synthetic reverberation. I think their test could have gone farther, as I will elaborate upon later, but nonetheless this first attempt, and the public release of the tools to build thereupon is laudable.

**Weaknesses:**

The authors claim in the abstract that the simulated reverberation is "highly realistic". They might be correct, but I find the evidence for this disappointingly thin. None of the issues I bring up here are cause for immediate rejection.  Ray-tracing algorithms have a tremendous role to play in acoustics research and this tool will likely be useful in many research projects. Moreover, given the dearth of large-scale datasets of reverberant IRs, perhaps limited models are better than none. However, so that other researchers can make good use this toolset, these limitations should be clearly stated in the main text of the paper.

The major issues I find are as follows:

- The direct comparisons between simulated and recorded reverberation are limited to only one room.  While it is certainly true that real-world IR measurements are not easy, 3 measurements each in 2 rooms, would have been better than 7 measurements in one room.  Moreover, a great many past research papers on reverberation have measured 3--10 rooms, and some many more.  Time-consuming is not the same as unfeasible, and if the authors want the research community to train algorithms with 10 million examples of simulated IRs, a comparison set of at least 10 rooms doesn't seem unreasonable. Given this, the authors should moderate their claim of "highly realistic" and highlight more thorough tests with real-world IRs as future work.

- The comparison between the recorded reverberation and the simulated is rather coarse. Both the RT60 and DRR are frequency dependent in real rooms. And large rooms exhibit prominent resonent peaks in the late tail due to resonant modes.  Do the soundscape 2.0 IRs capture these structures? If the answer is yes, than so much the better for soundscape 2.0.  If no, the discrepancies will be informative and useful to know.

- The simulation requires specifying a set of parameters for any material in the simulation scene (absorption, scattering, transmission). It is good that the authors have provided default values for common materials, and it is good that users can specify such values vi an API.  But I still have many questions: how were the default values obtained? From measurements? Or physics models? Or are they all obtained to optimize the fit to one single room as implied in footnote 7?  Moreover, I am curious to know how dramatically the reverberation would change if these values were altered. If the authors re-simulated the 7 IRs they compare with real-world recordings with the acoustic randomization procedure they applied in the speech recognition task, by how much will the simulated RT60 and DRR vary? If the answer is "not much, the RT60 and DRR are robust", then a skeptical reader will be reassured that the close match shown in Fig 3 is not just due to over-fitting parameters to a single room. If otherwise, readers will be forewarned that their own simulations may require some care and thought in the initial setup to yield high-quality audio. In which case, how should a user know how to adjust these parameters?

- Whereas graphics rendering can ignore the thickness of walls and surfaces, acoustic rendering does not have this luxury.  The absorption/transmission coefficients of real rooms will certainly be different for thin-vs-thick surfaces.  Presumably, this can be fixed by manually altering the material parameters, but this may require manually adjusting these values for any given rendered room. Again, future users of thie simulation tool should be forewarned that they may have to think about these issues to get high-quality results.

- In the supplemental material document the authors state, regarding Fig 4: "The main reason for the Matterport3D’s RT60 distribution skewing towards left is because there are lots of broken meshes in that dataset, which results in ray leaking from holes and smaller reverberation in general. On the contrary, Gibson and HM3D have higher quality mesh and have larger RT60s on average".  Of course the authors here cannot be blamed for problematic meshes.  A rendering algorithm can only be as good as the information fed into it.  But this does highlight an issue: standard 3D meshes are optimized for image rendering which faces different constraints than audio rendering (image rendering requires ray-tracers to track a much smaller number of reflections than audio renderers, so, holes in the mesh, which pose little problem for image rendering and might be common, can prove problematic for audio).  This implies that a fraction of the 10 million IRs in their public dataset underestimate the RT60 (how many?). The dataset is still of use, but this issue should be publicized.  Also, once more, those who wish to use the simulation tool should be forewarned they need to check their mesh quality.

- In the speech recognition task, the authors demonstrate that a speech recognition trained on reverberant speech performs better when trained on soundscape 2.0 reverberation rather than pyroomacoustics.  I assume they ensured that the distribution of RT60s and DRRs for the two sets of simulated IRs were matched, but this is not stated. In addition, I would be curious to see the effect if the speech recognition algorithm was trained with recorded reverberation from another dataset (Presumably, soundscape 2.0 would yield better performance because of the size of the simulated dataset), or with perceptually inspired statistical reverberation synthesis models, such as used in hearing research, or in music digital effects (Presumably, here soundscape 2.0 would yield better performance because the reverberation is more realistic). If this were demonstrated, it would provide evidence that soundscape 2.0 really is a state-of-the-art audio renderer. Until then, I would suggest the authors moderate their claims of "highly realistic", and mention these other tests as useful future work.

**Additional Feedback:**

N/A.

**Clarity:**

The paper is well written, except some key information is buried in the supplemental document, as I have detailed above in "weaknesses".


**Correctness:**

As listed in weaknesses, I think some of the claims made by the authors are not well supported.  But I do not see anything that seems obviously incorrect.


**Documentation:**

The documentation on the github page is quite good, in my view.

**Ethics:**

No concerns.

**Relation To Prior Work:**

The references are good and thorough. Some mention that digital music tools create compelling renditions of reverberation via statistical techniques might be warranted.  It suggests that there may be other approaches to reverberation synthesis beyond high-resolution and CPU-intensive ray-tracing.


**Summary And Contributions:**

The authors present a second-generation of the soundscape tool, which is designed to simulate realistic acoustic reverberation from 3-dimensional room/scene models, such as are already used for rendering computer graphics. Such a tool facilitates large-scale rendering of audio for simulated scenes. The authors have already run this simulation at scale and release a data set of 10-million pairs of rendered images with corresponding acoustic impulse-responses (IRs).

In addition to obvious uses in computer game audio rendering, such a tool may aid in the development of machine hearing algorithms, like speech recognition, which often show pathological behaviour in the presence of reverberation that humans would consider very moderate.

Of course, the use of such a tool, and associated simulated dataset, is predicated on the simulation accuracy (which, itself, relies on the accuracy of properties of the user generated scene model). To demonstrate the accuracy of soundscape 2.0 IRs, the authors show two demonstrations that appear successful, but in this reviewer's view, are not as strong as they could, or should, be.  This does not mean the work isn't useful, but the paper should certainly acknowledge the shortcomings directly and clearly.

Additionally, the authors present two additional benchmark tasks. In one, a simulated agent must localize a sound in a complex simulated environment. Here the others demonstrate that soundscape 2.0 improves upon the first generation soundscape in conveying (simulated) source location via (simulated) sound.  The other benchmark compares two soundscape 2.0 simulations of reverberation for one room, one made with "high-quality mode", and the other with "high-speed mode".  While these two benchmark tasks are not without merit, they do seem to belong in a different category than the tests that involve real-world audio.  I believe mixing all four tasks into one section on "Evaluation and benchmarks" obscures an important difference: the claims made with entirely simulated tasks are well justified, but those about accuracy are not.

The authors deserve great credit for tackling a relevant problem, and providing tools that allow audio rendering to be run at scale with pre-existing scene simulation tools, as is already common with image rendering. The synthetic reverberation might even prove to be accurate, however, the task of validating this is more difficult than suggested by this paper, and that should be addressed. Moreover, there is no reason to think high-quality reverberation can be simulated with a flawed 3D-scene model, no matter how good the simulation algorithm, and the paper (and documentation) should warn users of pitfalls.

---

> ### Author Response · Authors · 2022-08-29
> **Response to reviewer TAC1**
>
> We thank the reviewer for the encouraging feedback and insightful questions. We appreciate the reviewer's suggestion to take care to balance the strengths with clear acknowledgement of certain shortcomings (e.g., assumptions about input mesh quality, the limits of RT60 and DRR as metrics, practical challenges in specifying material properties), and we have revised our text accordingly.
>
> **Q1: Comparisons with real IRs limited to one room.**
> We choose to collect acoustic measurements in the Replica apartment (which contains multiple rooms, albeit with an open floorplan) for several reasons: 1. the apartment is representative of real spaces (proper decorations and furniture instead of empty space used in many acoustic labs) 2. the apartment has complex room geometry instead of simple shoebox room geometry 3. the mesh of the environment has been scanned and shared publicly in the Replica dataset. This makes it possible for others to simulate their IRs using the same mesh environment and compare with our simulated as well as collected IRs. For other rooms in the Replica dataset or other scene datasets (e.g., Matterport3D), we generally don't have physical access to them and thus can't collect acoustic measurements in these spaces.
>
> SoundSpaces 2.0 is applicable to many different meshes (HM3D, Gibson, etc.), as shown in Section 3.4. We agree that future work could record real IR measurements in other diverse environments, and have added this in Section 6.
>
> **Q2: Coarse comparison between recorded and simulated IRs.**
> In Supp. material, beyond the RT60 and DRR, we provide the measured and simulated IRs to enable more detailed analysis and comparison. We limited the comparison to the 250-4kHz band due to measurement equipment frequency response limitations, and because the comparison was less sensitive to measurement imperfections. As for resonant modes, they are not captured by our path tracer (or any path tracer in general), which is a well known shortcoming of geometrical-acoustic modeling techniques. We have acknowledged this shortcoming in Section 6.
>
> **Q3: Questions about common materials.**
> The common acoustic materials were obtained from an acoustic database [1], where acoustic properties are measured for common materials. For the real IR comparison, if we vary the material properties in the same way as in the acoustic randomization strategy, we obtain IRs with similar RT60 and DRR, which shows the close match in Fig 3 is not just due to over-fitting parameters in the same room.
>
> **Q4: Impact of the thickness of walls and surfaces on material properties.**
> We model distance-dependent damping of the sound propagation media (L205-L208) and thus the thickness of walls and surfaces is taken in the account in the propagation engine. This damping is in addition to the attenuation modeled by the material transmission coefficients.
>
> **Q5: Impact of broken meshes on rendering quality.**
> This is a great point. To further assist users in creating high-quality acoustic rendering, we have added an API for checking the amount of rays leaked from broken meshes.  In this way, users could check the quality of their mesh for acoustic rendering and they can fix the mesh accordingly. We will re-render the PanoIR dataset with IRs that have high ray efficiency (low ray leakage). In addition, we have pushed the discussion on mesh quality from Supp.~to main (Section 6) to make readers aware of this important factor.
>
> **Q6: Comparison with real IR on finetuning ASR model.**
> Correct, for simulated IRs, we ensure the distributions of RT60s and DRRs are matching.  We have made this explicit in L321. As requested, we conducted an additional experiment where the ASR model is finetuned on a real RIR collection [2], which consists of IRs that come from three datasets: the RWCP sound scene database, the 2014 REVERB challenge database, and the Aachen impulse response database (AIR). This baseline has a test WER of 13.32\%, outperforming the Pyroomacoustics (16.24\%) but underperforming the model finetuned on IRs generated on our SoundSpaces 2.0 platform (12.48\%). This reinforces our claim about the realism of SoundSpaces for the speech recognition experiment.
>
> [1] Architectural Acoustics, M. David Egan, McGraw-Hill, 1988 - Acoustical engineering.
>
> [2] A study on data augmentation of reverberant speech for robust speech recognition, Tom Ko et al., ICASSP 2017

---

### Official Review · Reviewer_hbPy · 2022-07-29
**A platform for audio-visual simulation**

**Rating:** 7
**Confidence:** 2
**Correctness:** Probably correct

**Strengths:**

1. The platform can be very useful for the research community as it supports audio-visual rendering and can be used in any environment.
2. The platform supports acoustic and spatial continuity and has multiple configurable parameters like frequency bands, diffraction, e.t.c
3. The authors have shown results for multiple down-stram tasks such as far-field speech recognition.

**Weaknesses:**

1. The baseline could have been improved.
2. The authors addressed improving the accuracy of the direct-to-reverent ratio (DRR) by fixing the bias of √4ℼ present in the original Soundspaces. However, no discussion is presented on this in the paper.
3. There is no discussion on the impact of reducing the number of rays in high-speed mode. How does the reduction in the number of rays affect the performance?
4. The sub-section 5.2 compares the simulation accuracy with Real IRs using the FRL apartment from the Replica dataset. The comparison should be made in more diverse settings (like crowdy apartments, environmental location).

**Additional Feedback:**

No

**Clarity:**

Lack of clarity at multiple places. It could have been written better. There are a few typos, such as line no 49.

**Documentation:**

Yes

**Ethics:**

Nothing is mentioned about the ethics part

**Relation To Prior Work:**

Yes

**Summary And Contributions:**

The paper presents a platform for audio-visual simulation that can be used for audio-visual rendering. The paper also presents a dataset called Sound-spaces-PanoIR for acoustic-visual learning tasks. The simulation platform has multiple configurable parameters.

---

> ### Author Response · Authors · 2022-08-29
> **Response to reviewer hbPy**
>
> We thank the reviewer for the encouraging feedback and insightful questions.
>
> **Q1: The baseline could have been improved**
> We are not clear what baseline the reviewer is referring to or what should be improved; we would welcome further detail.
>
> **Q2: Discussion on DRR improvement.**
> The original SoundSpaces wrongly assumed the listener having a built-in attenuation of $\sqrt{4\pi}$, which produces an incorrect reverb level.  We fixed this bias (see L148).
>
> **Q3: Discussion on the impact of reducing the number of rays.**
> Sec. 5.1 discusses the tradeoff between simulation speed and quality.  We show that high-speed mode greatly improves efficiency over the high-quality mode, by 8X with single thread and 33X with 5 threads, while only losing 9.5\% accuracy despite RT60 calculation being noisy. In high-speed mode, if we keep reducing the number of rays, the aforementioned trend still holds true: reducing the number of rays leads to faster but less accurate simulation. With the ability to control simulation parameters, users could further tweak the simulation beyond the two recommended modes to find the setting that works the best for their task and mesh environment.
>
> **Q4: Real comparison in a more diverse settings.**
> We choose to collect acoustic measurements in the Replica apartment for several reasons: 1. the apartment is representative of real spaces (proper decorations and furniture instead of empty space used in many acoustic labs) 2. the apartment has complex room geometry instead of simple shoebox room geometry 3. the mesh of the apartment has been scanned and shared in the Replica dataset [1]. This makes it possible for others to simulate their IRs using the same mesh environment and compare with our simulated as well as collected IRs. For other rooms in the Replica dataset or other scene datasets (e.g., Matterport3D), we generally don't have physical access to them and thus can't collect acoustic measurements in these spaces.
>
> We note that SoundSpaces 2.0 is already extensible to many different meshes (HM3D, Gibson, etc.), as shown in Section 3.4. Future work could build upon this work to expand the real IR measurement collection to other diverse environments.
>
> [1] The Replica Dataset: A Digital Replica of Indoor Spaces, Julian Straub et al., arXiv 2019

---

### Author Response · Authors · 2022-08-29
**Overall response**

All six reviewers recommend accepting the paper (all scores are 7: Good paper, accept). We thank the reviewers for their valuable feedback and suggestions. All reviewers find that our platform is very helpful for the research community and opens up opportunities for embodied AI, multi-modal perception, and audio research.

With the additional page allowed in rebuttal, we incorporated the feedback from reviewers into our paper (revised text shown in blue). The main changes include adding two additional baselines in Table 4 (not affecting our initial conclusion), pushing discussion on limitations from Supp. to the main, and discussing speed difference with other platforms in Section 5.1. Note that line numbers and section numbers in this rebuttal refer to the updated pdf available on OpenReview.

In addition, since the release of this platform two months ago, 15+ researchers across different universities have started to use our platform and based on their feedback, we added more important features and functionalities, including supporting multiple receivers/listeners directly, creating and simulating shoebox rooms with different materials for walls, loading custom head-related transfer functions (HRTFs), and an API for checking the amount of rays leaked during simulation. These updates are reflected in the documentation as well as in the paper.

Next, we respond to each reviewer individually.

---

### Public Comment · ~Andrew_Adams3 · 2022-09-27
**Thanks**

Thanks. Yesterday, I was looking for the best [dissertation proofreading services uk](https://www.uk-dissertation.com/proofreading.html) online, but I wasn't sure which one to choose since there were so many options available. Fortunately, one of my friends gave me the name of a website that will be very useful for me as I finish the chapters of my dissertation.

---

### Meta-Review · Area_Chair_vVKY · 2022-09-10

**Recommendation:** Accept
**Confidence:** 5

**Metareview:**

This paper was reviewed by six experts and received all positive scores. AC feels this work is interesting and deserves to be published on NeurIPS 2022 dataset track. The reviewers did raise some valuable concerns that should be addressed in the final camera-ready version of the paper. The authors are encouraged to make the necessary changes in the final version.

---

### Decision · Program_Chairs · 2022-09-16

Accept